

# Dietary supplementation of black soldier fly (*Hermetica illucens*) meal modulates gut microbiota, innate immune response and health status of marron (*Cherax cainii*, Austin 2002) fed poultry-by-product and fishmeal based diets

Md Javed Foysal[1,2], Ravi Fotedar[1], Chin-Yen Tay[3] and
Sanjay Kumar Gupta[1,4]

[1] School of Molecular and Life Sciences, Curtin University, Bentley, WA, Australia
[2] Department of Genetic Engineering and Biotechnology, Shahjalal University of Science & Technology, Sylhet, Bangladesh
[3] Helicobacter Research Laboratory, Marshall Centre for Infectious Disease Research and Training, School of Biomedical Sciences, University of Western Australia, Perth, WA, Australia
[4] ICAR-Indian Institute of Agricultural Biotechnology, Ranchi, Jharkhand, India

Corresponding authors
Md Javed Foysal,
mdjaved.foysal@postgrad.curtin.
edu.au
Sanjay Kumar Gupta,
sanfish111@gmail.com

## ABSTRACT

The present study aimed to evaluate the dietary supplementary effects of black soldier fly (*Hermetia illucens*) (BSF) meal on the bacterial communities in the distal gut, immune response and growth of freshwater crayfish, marron (*Cherax cainii*) fed poultry-by-product meal (PBM) as an alternative protein source to fish meal (FM). A total of 64 marron were randomly distributed into 16 different tanks with a density of four marron per tank. After acclimation, a 60-days feeding trial was conducted on marron fed isonitrogenouts and isocalorific diets containing protein source from FM, PBM, and a combination of FM + BSF and PBM + BSF. At the end of the trial, weight gain and growth of marron were found independent of any dietary treatment, however, the two diets supplemented with BSF significantly ($P < 0.05$) enhanced haemolymph osmolality, lysozyme activity, total haemocyte counts, and protein and energy contents in the tail muscle. In addition, the analysis of microbiota and its predicted metabolic pathways via 16s rRNA revealed a significantly ($P < 0.05$) higher bacterial activity and gene function correlated to biosynthesis of protein, energy and secondary metabolites in PBM + BSF than other dietary groups. Diets FM + BSF and PBM + BSF were seen to be associated with an up-regulation of cytokine genes in the intestinal tissue of marron. Overall, PBM + BSF diet proved to be a superior diet in terms of improved health status, gut microbiota and up-regulated expression of cytokine genes for marron culture.

## INTRODUCTION

Fish meal (FM) is one of the major sources of dietary protein for cultured aquatic animals, posing an increasing challenge on the reduction of feed cost and wild fish stocks (*Tacon & Metian, 2008*). Recent reports revealed the global decline of FM production due to the increasing demand of wild stocks in contrast to the amount of fish harvested for trashing (*Cashion et al., 2017*; *Pauly & Zeller, 2017*). It is an utmost priority to find a suitable, cheap and widely available alternative to FM to meet-up the burgeoning demand of feed for the sustainable aquaculture industry. On the other hand, poultry-by-products meal (PBM) is a solidified by-product from poultry processing industry that has high potential to be incorporated into aqua-diets as a substitute for FM (*Saoud et al., 2008*). PBM is readily available worldwide and is more economical when compared with FM (*Emre, Sevgili & Diler, 2004*; *Wu et al., 2018*). PBM alone or in combination with bone, feather, and blood meal, can be one of the major protein source in aqua-diets owing to high protein, suitable fatty acids, vitamins, and minerals, (*Dozier, Dale & Dove, 2003*; *Fuertes et al., 2013*). In addition, PBM is reported to have a positive effect on the growth rate, digestibility, and immune status of fish and crayfish (*Bransden, Carter & Nowak, 2001*; *Yang et al., 2006*; *Saoud et al., 2008*). The black soldier fly (BSF) larvae, another alternative to FM protein source, is gaining increasing popularity for the aquaculture industry (*Stamer et al., 2014*). BSF contains high protein and fat, rich in trace elements, and more importantly, has a lesser impact on the environment and hence named as "savior" in the food-insecure world (*Wang & Shelomi, 2017*). The efficacy of BSF as an alternative protein source for warm-water fish has been validated in earlier studies (*Stankus, 2013*; *Stamer et al., 2014*; *Wang & Shelomi, 2017*); however, limited data are currently available on the suitability of using BSF as alternative protein source and its supplementary effect on other sources of protein, including FM and PBM in aqua-diets (*St-Hilaire et al., 2007*; *Kroeckel et al., 2012*).

Marron (*Cherax cainaii*) is one of the largest freshwater crayfish endemic to the southern part of Western Australia (WA) (*DoF, 2010*; *Foysal et al., 2019b*). Marron is popular for its distinctive flavor and taste and therefore considered as an iconic freshwater crayfish in WA. Marron farming possesses a number of advantages over other crayfish species (*C. destructor*, *C. preissii*, and *C. crassimanus*) including disease resistance, high market demand and price, and possible live shipment to long distances etc., (*Lawrence, 2007*). Yet, the production has remained stagnant over the decade (*Van Mai & Fotedar, 2018*). To improve the production potential, several investigations have been conducted to find the suitable combination of feed for marron (*Ambas, Suriawan & Fotedar, 2013*; *Ambas & Fotedar, 2015*; *Ambas, Buller & Fotedar, 2015*). As an omnivorous animal, the protein requirement for marron is moderate (approximately 30%) (*Ambas, Fotedar & Buller, 2017*). Therefore, a combination of PBM and BSF can be an ideal alternative protein source in marron diet.

Crayfish harbor complex bacterial communities in the gut that influences various host functions like digestion, nutrition, immunity, and disease resistance (*Skelton et al., 2017*; *Zoqratt et al., 2018*). Due to their vital roles, enrichment of some bacterial communities in the gut with dietary feed supplementation has been validated to accelerate

the growth and immune status of the crayfish (*Anuta et al., 2011*; *Safari & Paolucci, 2018*). High relative abundance of lactic acid bacteria (LAB), especially *Bacillus*, *Lactobacillus* in the gut are described to have beneficial influences on the health and immunity of white shrimp (*Liptopenaeus vannamei*) (*Zokaeifar et al., 2012*), zebrafish (*Danio rerio*) (*Falcinelli et al., 2015*), Atlantic salmon (*Salmo salar*) (*Gajardo et al., 2017*), and northern snakehead (*Channa argus*) (*Miao et al., 2018*).

Recent advancement in 16S rRNA based high throughput sequencing techniques has enabled precise detection of gut microbial communities of fish. This technology has been successfully employed to analyze the feeding effects on the gut bacterial populations of fish in earlier reports (*Gajardo et al., 2017*; *Miao et al., 2018*). In addition, quantitative real-time PCR (qRT-PCR) is routinely used for the measurement of the relative expression level of immune genes to infer the effect of feed on the immunity of crayfish (*Shekhar, Kiruthika & Ponniah, 2013*; *Jiang et al., 2015a*). This study aimed to characterize the bacterial communities in the distal gut of marron fed FM, PBM, a combination of FM + BSF and PBM + BSF and analyze the expression patterns of some cytokine genes.

## MATERIALS AND METHODS

### Experiment set-up

A total of 64 marron (65.01 ± 5.09 g) were procured from Blue Ridge Marron, Manjimup, WA (34.2019°S, 116.0170°E) and transported live to Curtin University Aquatic Research Laboratory, Technology Park, Bentley, WA. Marron were then distributed into 16 different tanks (80 cm diameter and 50 cm height, 250 L capacity) filled with 150 L of fresh water and acclimated for 14 days before commencement of the trial. Constant aeration was provided by air diffusers (Aqua One, Perth, WA, Australia) and fixed temperature (22 ± 0.5 °C) was maintained using submersible thermostat set (Aqua One, Perth, WA, Australia). Water quality parameters including pH, dissolved oxygen (DO), nitrate, nitrate, and ammonia were maintained within the suitable range for marron culture as reported previously (*Ambas, Fotedar & Buller, 2017*) (Dataset S1). The pH and temperature of the water were measured using portable waterproof °C/mV/pH meter (CyberScan pH 300; Eutech Instruments, Ayer Rajah Crescent, Singapore) while DO concentration was measured using digital DO meter (YSI55; Perth Scientific, Malaga, WA, Australia). The concentration of nitrate, nitrite and ammonia in water were monitored using Hach DR/890 Colorimeter (Hach, Loveland, CO, USA). Approximately 30% of water from each tank was exchanged twice a week after siphoning out the faecal wastes. Each marron was nurtured in a separate cage prepared of plastic mesh (0.8–8.0 mm thickness) to avoid cannibalism (*Ambas, Fotedar & Buller, 2017*). During acclimation, marron were fed every day at afternoon with standard basal marron diet composed of 29% crude protein, 9% crude fat, 8.5% moisture, and 5% crude ash, prepared by Glenn Forest, WA.

### Test diets and the experimental design

Four isoproteic and isocalorific diets containing FM, PBM, fishmeal supplemented with black soldier fly meal (FM + BSF), and PBM supplemented with black soldier fly meal (PBM + BSF) were prepared (Table 1). The ingredients were supplied by Glenn Forrest,
**Table 1 Feed ingredients and proximate composition (% dry weight) of the marron diets used in this study.**

| Ingredients[*] | FM | PBM | FM+BSF | PBM + BSF |
|---|---|---|---|---|
| Fishmeal | 41 | 0 | 32 | 0 |
| Poultry by product meal | 0 | 39 | 0 | 31.5 |
| Black soldier fly meal | 0 | 0 | 12 | 11 |
| Soya bean meal | 10 | 10 | 10 | 10 |
| Wheat | 37 | 38 | 36 | 36 |
| Corn starch | 4.80 | 4.80 | 4 | 4.80 |
| Cod liver oil | 4.20 | 5.20 | 3 | 3.70 |
| $CaCO_3$ | 0.02 | 0.02 | 0.02 | 0.02 |
| Vitamin premix | 0.23 | 0.23 | 0.23 | 0.23 |
| Vitamin C | 0.05 | 0.05 | 0.05 | 0.05 |
| Cholesterol | 0.50 | 0.50 | 0.50 | 0.50 |
| Lecithin-Soy | 1 | 1 | 1 | 1 |
| Betacaine | 1.20 | 1.20 | 1.20 | 1.20 |
| Total | 100 | 100 | 100 | 100 |
| Proximate composition of (g $kg^{-1}$ dry weight basis) | | | | |
| CP% | 29.93 | 29.61 | 30.07 | 30.20 |
| Lipid% | 7.12 | 7.32 | 7.56 | 7.48 |
| GE MJ $kg^{-1}$ | 18.21 | 18.75 | 18.45 | 18.53 |

Notes:
FM, fishmeal; PBM, poultry-by-product meal; FM + BSF, fishmeal + black soldier fly meal; PBM + BSF, poultry-by-product meal + black soldier fly meal; CP, crude protein; GE, gross energy; MJ, mega joule.
[*] All ingredients were procured and diets were prepared by Glenn Forest Specialty Feeds, Western Australia.

WA and after feed formulation the test diets were also prepared by the same company. Proximate compositions of each diet was determined as per the method of *Association of Official Agricultural Chemists (AOAC) (2006)*. Four randomly assigned tanks, with four individually held marron in cages, were given each test diet, hence using a design of four dietary treatments × four replicates per dietary treatment. Each marron in a cage was fed the respective diets, once every day at 12 PM.

## Marron sampling

For haemolymph, health indices and gene expression analysis, 16 marron, one marron randomly selected from each tank was used. However, for DNA extraction, nine randomly selected marron from three randomly selected tanks from each treatment, were selected. No samples were collected from one of the tanks from each treatment.

## Analysis of growth, hemolymph parameters and health indices

At the end of the experiment, the marron growth was measured by using the following formulas: Weight gain (WG) = ((mean final body weight − mean initial body weight))/ (mean initial body weight); and Specific growth rate, (%/day) = ((mean final body weight − in mean initial body weight)/number of days) × 100.

The haemolymph osmolality (HO) was measured (*Sang & Fotedar, 2004*) at 96 h intervals. Haemolymph (0.05 mL) from each marron was extracted carefully from

the pericardial cavity using 0.5 mL syringe containing a 0.1 mL of precooled anticoagulant (0.1% glutaraldehyde in 0.2M sodium cacpodylate, pH 7.0 ± 0.2). The osmolality of the hemolymph and anticoagulant mix solution was measured by Cryoscopic Osmometer-Osmomet 030 (Gonotec, Berlin, Germany).

Similarly, the haemolymphatic lysozyme activity was measured using turbid-metric assay, as described by *Van Mai & Fotedar (2018)*. A total of 50 µL of anticoagulant added hemolymph sample were dispersed into 96-well microtiter-plate (Iwaki, Tokyo, Japan). After 15 min incubation, 50 µL of PBS (0.25 mg/mL) suspended *Micrococcus lysodeiktikus* (Sigma-Aldrich, St. Louis, MO, USA) solution was added into separate wells of the same plate. The absorbance of each well in the plate was then monitored at 2 min intervals for 20 min (10 times) at 450 nm wavelength in MS212 reader (Titertek Plus; Tecan, Grodig, Austria). The lysozyme activity unit was defined as the amount of enzyme consumed in a reduction absorbance of 0.0001/min. The lysozyme activity was expressed as enzyme units/mL of haemolymph (U/mL).

Similar to the above parameters, to measure total haemocyte counts (THC), 0.2 mL of marron haemolymph was suspended in 0.2 mL of anticoagulants. THC was calculated for each marron under a hemocytometer (Nauabuer, Darmstadt, Germany) with 100× magnification (*Ambas, Fotedar & Buller, 2017*).

The amount of protein and energy in the tail muscle and crude fat in the hepatopancreas of marron was measured using methods described by the *Association of Official Agricultural Chemists (AOAC) (2006)*. Crude protein was measured following the Kjeldahl method ($N \times 6.25$) using sulfuric acid ($H_2SO_4$) and copper catalyst tablets in Kjeltec Auto 1030 analyzer (Foss Tecator, Höganäs, Sweden). The percentage of crude fat in marron hepatopancreas was calculated using Soxhlet ether extraction method using Soxtec System HT6 (Tecator, Höganäs, Sweden), as described earlier (*Jim, Garamumhango & Musara, 2017*). The total gross energy in the dried tail muscle content was determined by using bomb calorimeter (Heitersheim, Germany). The hepatopancreas index (HI) of marron after feeding trial was calculated using the formula:

Hepatopancreas index
= Hepatopancreas dry weight/(body dry weight excluding large cheliped × 100).

## DNA extraction, PCR, and high throughput sequencing

At the end of the experiment, randomly selected marron were taken into biosafety cabinet followed by careful excision of the guts. Then the hindgut were separated and the gut contents of three marron from each tank were homogenized, pooled together, and transferred into 1.5 mL of Eppendorf. The bacterial DNA from the hindgut sample was extracted using DNeasy Blood and Tissue Kit (Qiagen, Crawley, UK) following the manufacturer's instructions. Extracted DNA was quantified using NanoDrop spectrophotometer (Thermo Fisher Scientific, Waltham, MA, USA) and diluted eventually to 30 ng/µL final concentration. PCR master mixture was prepared as 50 µL final concentration containing 25 µL Hot Start 2X Master Mix (New England BioLabs Inc., Ipswich, MA, USA), two µL of template DNA, one µL of each forward and

reverse V3–V4 sequencing primers, and 21 μL of nuclease-free water. The PCR was performed using BioRad S100 Gradient Thermal Cycler (Bio-Rad Laboratories, Inc., Foster City, CA, USA), under the following conditions: 2 min initial denaturation at 94 °C; 30 cycles of denaturation (30 s at 94 °C), annealing (1 min at 55 °C), and extension (1 min at 68 °C); a final extension at 68 °C for 5 min and holding the temperature at 10 °C. The primers set, obtained from Illumina protocol (Part # 15044223 Rev. B) 16sF 5′-TCGTCGGCAGCGTCAGATGTGTATAAGAGACAGCCTACGGGNGGCWGCAG and 16sR 5′-GTCTCGTGGGCTCGGAGATGTGTATAAGAGACAGGACT ACHVGGGTATCTAATCC were used to amplify the 16s V3–V4 region. Amplified PCR products were separated by 1% agarose gel electrophoresis (BioRad Laboratories Inc., Hercules, CA, USA) and visualized under gel doc (FujiFilm LAS–4000 Image Analyzer, Boston Inc., Foster City, CA, USA). The 16S rRNA PCR amplicon of each sample was barcoded via a secondary PCR according to the Illumina standard protocol (Part # 15044223 Rev. B). Each sample was sequenced up to 50,000 reads on an Illumina MiSeq platforms (Illumina Inc., San Diego, CA, USA) at Harry Perkins Institute of Medical Research, WA, using Illumina MiSeq Reagent v3 kit (600 cycles, Part # MS-102-3003).

## Quantitative real-time PCR

In this study, five cytokine genes (interleukin (IL)-1β, IL-8, IL-10, IL-17F, and tumour necrosis factor (TNF)-α) associated with immunity of fish and crayfish (*Miao et al., 2018*), and two reported crayfish tissue genes, vitellogenin (*Vg*) and proliferating cell nuclear antigen (*Pcna*) (*Jiang et al., 2015b*) were selected for qRT-PCR to measure the relative expression patterns after feeding trial. The whole intestine tissue samples were collected aseptically, chopped into small pieces, and preserved into RNA Later solution (Sigma-Aldrich, Darmstadt, Germany) according to manufacturer's instructions. After thawing and drying, samples were homogenized and ground into a fine powder. Five milligrams of homogenized tissue samples were used for RNA extraction using RNeasy Mini Kit (Qiagen, Hilden, Germany). During extraction DNase-I (Qiagen, Hilden, Germany) was added to remove DNA impurities. Extracted RNA was checked for quality and quantity in 1% agarose gel and NanoDrop spectrophotometer, respectively (Thermo Fisher Scientific, Waltham, MA, USA). The cDNA library from the extracted RNA was prepared using Omnicript RT kit (Qiagen, Hilden, Germany). qRT-PCR to measure the relative expression level of eight genes was performed using PowerUp™ Cyber Green Master Mix (Thermo Fisher Scientific, Waltham, MA, USA) with 7500 Real-Time PCR System (Applied Biosystems, Foster City, CA, USA). The relative expression of each target gene was calculated using the $2^{-\Delta\Delta CT}$ method, after normalization against the β-actin reference gene (*Miao et al., 2018*). Duncan's multiple range test was used to compare the relative expression level of genes for FM to each of the others (PBM, FM + BSF, and PBM + BSF).

## Bioinformatics

The initial quality of 16S rRNA sequences was checked in fastQC pipelines (*Andrews, 2010*). The sickle program was used for quality trimming of sequencing reads. Following trimming, reads of less than 200 bp were discarded (*Joshi & Fass, 2011*). MeFiT pipeline
**Table 2 Water quality parameters in marron tanks fed different proteins.**

| Parameters | FM | PBM | FM + BSF | PBM + BSF |
|---|---|---|---|---|
| Temperature (°C) | 22.3 ± 0.014 | 22.3 ± 0.011 | 22.4 ± 0.013 | 22.4 ± 0.01 |
| pH | 7.5 ± 0.006 | 7.5 ± 0.005 | 7.5 ± 0.005 | 7.6 ± 0.003 |
| DO (mg/L) | 6.51 ± 0.003 | 6.55 ± 0.003 | 6.31 ± 0.006 | 6.65 ± 0.002 |
| Nitrate (mg/L) | 0.049 ± 0.001 | 0.047 ± 0.001 | 0.048 ± 0.001 | 0.046 ± 0.001 |
| Nitrite (mg/L) | 0.015 ± 0.0005 | 0.017 ± 0.0007 | 0.016 ± 0.0006 | 0.018 ± 0.0008 |
| Ammonia (mg/L) | 0.20 ± 0.01 | 0.22 ± 0.01 | 0.21 ± 0.01 | 0.22 ± 0.01 |

Note:
DO, dissolved oxygen; FM, fishmeal; PBM, poultry-by-product meal; FM + BSF, fishmeal + black soldier fly meal; PBM + BSF, poultry-by-product meal + black soldier fly meal.

was then used for the merging of overlapping paired-end reads with default parameters (*Parikh et al., 2016*). Filtering of chimeric sequences, open reference clustering of 16S rRNA sequences into operational taxonomic units (OTUs) at 97% similarity threshold and removal of singleton OTUs were conducted using micca otu (version 1.7.0) (*Albanese et al., 2015*). Taxonomic assignment of the representative OTUs was performed using mica classify against SILVA 1.32 database clustered at 97% identity (*Quast et al., 2013*). Multiple sequence alignment of the representative OTUs was done using PASTA algorithm (*Mirarab et al., 2015*). Non-metric multidimensional scaling of bacterial OTUs from four different groups was done in PAST statistical software package (*Hammer, Harper & Ryan, 2009*). The Shannon, Simpson, and Fisher alpha diversity indices were calculated using "vegan" package in R (*Oksanen et al., 2018*). Chao1 diversity in the samples was calculated using formula $S_{chao1} = S_{obs} + (n1)^{\surd}/2n2$, where $S_{obs}$ = number of observed genera, $n1$ = number of singletons (genus captured once), $n2$ = number of doubletons (genera captured twice) (*Militon et al., 2010*). Venn diagram for bacterial abundance regarding diversity at genus level was generated using FunRich (v3.1.3) (*Benito-Martin & Peinado, 2015*). Linear Discriminant Analysis Effect Size, (LEfSe) (University of Auckland, Auckland, New Zealand) was applied to find the indicator bacterial groups in four different feeding groups with a minimum logarithmic LDA cut-off value of 8.0. For predicting differentially abundant metabolic pathway in four different groups, Piphillin algorithm (http://secondgenome.com/Piphillin) was used with supports of KEGG database, BioCyc (v21), and LEfSe (LDA 3.0) (*Iwai et al., 2016*). Numerical growth and health indices data for marron were analyzed using SPSS IBM (v23, 2017). One way analysis of variance (ANOVA) was used to calculate any significant differences ($P < 0.05$) among variables in four different dietary treatments.

## RESULTS

### Effects of four different feed on marron health indices

No significant ($P > 0.05$) WG and growth rate were observed in PBM and two BSF supplemented diet (FM + BSF and PBM + BSF) compared to FM (Table 2; Dataset S2). The effects of dietary PBM was found to be almost similar to FM on marron health and immunity. Both BSF (FM + BSF and PBM + BSF) supplemented diets showed significant enhancement in protein, energy, lysozyme, and THC. However, the diet

**Table 3  Mean ± SE of some health parameters of marron at the end of feeding trial.**

| Parameters | FM | PBM | FM + BSF | PBM + BSF |
|---|---|---|---|---|
| WG | 24.33 ± 1.38 | 23.91 ± 0.77 | 25.35 ± 0.85 | 26.49 ± 0.84 |
| SGR | 0.51 ± 0.26 | 0.49 ± 0.12 | 0.51 ± 0.13 | 0.53 ± 0.14 |
| Protein[†] | 82.4 ± 0.67 | 83.2 ± 0.56 | 85.4 ± 1.01* | 89.4 ± 0.95** |
| Energy[†] | 20,010.3 ± 75.11 | 20,242.3 ± 89.24 | 21,672.8 ± 311.5* | 21,778.5 ± 356.9* |
| Fat[¶] | 8.9 ± 0.11 | 8.4 ± 0.11 | 6.4 ± 0.50* | 6.1 ± 0.40* |
| HI | 38.5 ± 0.52 | 36.9 ± 0.27 | 34.5 ± 0.31* | 32.2 ± 0.72* |
| Lysozyme | 0.42 ± 0.006 | 0.4 ± 0.003 | 0.47 ± 0.011* | 0.59 ± 0.036** |
| THC | 8.0 ± 0.09 | 7.9 ± 0.09 | 8.2 ± 0.13* | 9.5 ± 0.15** |
| HO | 0.4 ± 0.003 | 0.41 ± 0.005 | 0.42 ± 0.011 | 0.46 ± 0.010** |

Notes:
FM, fishmeal; PBM, poultry-by-product meal; FM + BSF, fishmeal + black soldier fly meal; PBM + BSF, poultry-by-product meal + black soldier fly meal; WG, weight gain; SGR, specific growth rate; HI, hepatopancreas index; THC, total haemocyte count; HO, haemolymph osmolality.
* Significantly different at α-level of 0.05.
** Significantly different at α-level of 0.005.
† From tail muscle.
¶ From hepatopancreas.

PBM + BSF showed a significant ($P < 0.05$) influence on the energy content in tail muscle, and significantly pronounced ($P < 0.005$) effects of PBM + BSF was recorded for tail muscle protein, lysozyme, THC and HO (Table 3; Dataset S3). Dietary incorporation of BSF with PBM also showed a significant ($P < 0.05$) decline in HI and fat content of hepatopancreas whereas both BSF (FM + BSF and PBM + BSF) supplemented diets showed significant enhancement in protein, energy, lysozyme, and THC.

## Bacterial diversity in the hindgut after feeding trial

At the end of the trial, all the four experimental diets displayed distinct effect on different bacterial populations of marron hindgut (Fig. 1A). Quality filtering obtained 377,848 high-quality reads which were assigned into 119 genera. The number of the shared and unique genera were found to be significantly ($P < 0.05$) higher in PBM + BSF fed marron than the other marron groups (Fig. 1B). At genus level, *Proteobacteria* (46.62–92.88%) was found to be dominant in FM, PBM, and FM + BSF dietary groups whereas significantly ($P < 0.05$) higher abundance for *Firmicutes* (63.52–92.88%) was recorded in PBM + BSF fed group (Fig. 2A). Next to *Proteobacteria*, *Fusobacteria* (36.74–39.15%) and *Tenericutes* (20.28–39.77%) were the second abundant phyla in FM and FM + BSF fed marron, respectively. In the PBM + BSF fed marron, *Fimicutes* profusion (25.18–42.37%) was recorded, followed by *Proteobacteria* (28.4–33.6%), and *Bacteroidetes* (10.54–11.8%). Compared to FM, the other diets showed a significant ($P < 0.05$) increase in the diversity of bacteria at genus level whereas, the BSF supplemented diet, the marron displayed higher abundance. At genus level, *Hypnocyclicus* (46.31–47.81%), *Aeromonas* (39.93–58.5%), *Candidatus Bacilloplasma* (27.88–40.65%), and *Streptococcus* (17.77–24.38%), were the most dominant bacteria in FM, PBM, FM + BSF, and PBM + BSF fed marron, respectively (Fig. 2B). The relative abundance level for *Vibrio* was found to decrease significantly in PBM + BSF (1.13–2.42%) groups than FM (16.6–38.9%) fed group.

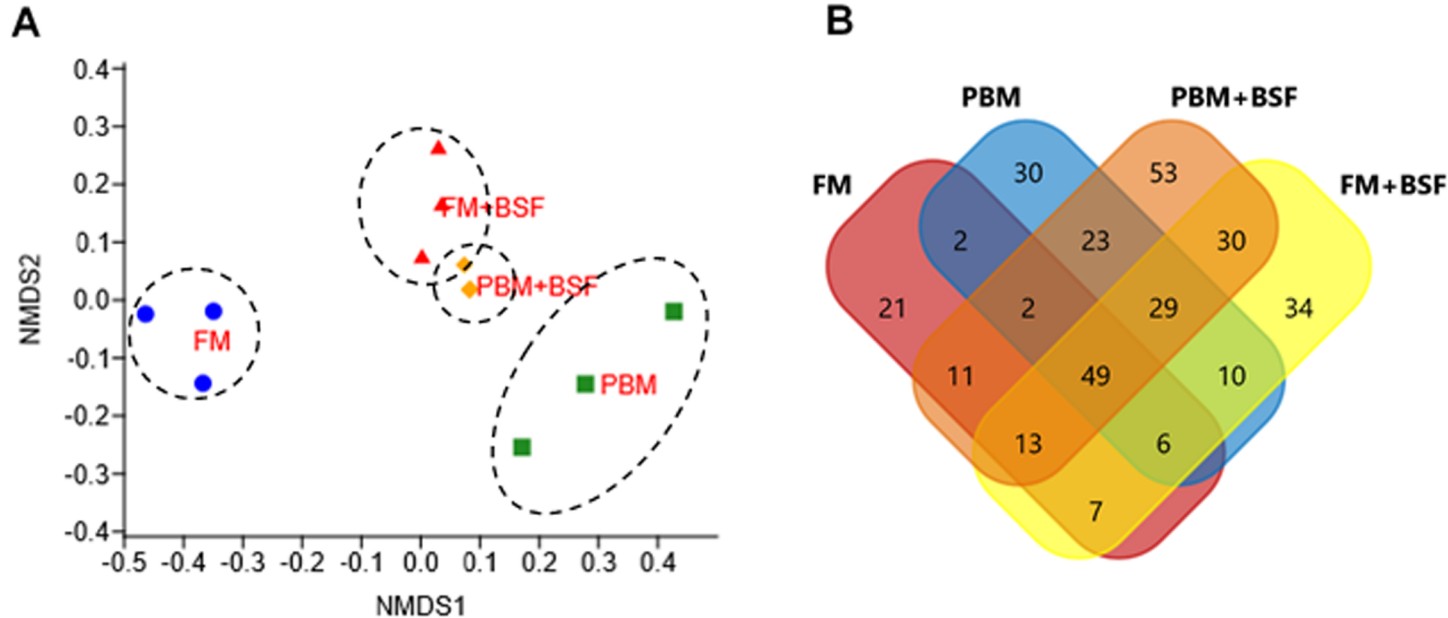

**Figure 1** **Effects of four different dietary protein on marron distal gut.** (A) Non-metric multidimensional scaling (nMDS) plot showing feeding effects on microbiota of distal gut in marron. (B) Number of shared and unique genera in four different protein fed groups. Abbreviation: FM, Fishmeal; PBM, Poultry-by-product meal; FM + BSF, Fishmeal + Black soldier fly meal; PBM + BSF, Poultry-by-product meal + Black soldier fly meal.

The abundance of *Serratia* and *Enterobacter* dropped to zero in PBM + BSF fed marron whereas, FM fed marron had *Serratia* and *Enterobacter* in abundance of 2.99–5.0% and 1.58–2.57%, respectively. Among the top 20 abundant genera, *Polynucleobacter*, *Limnohabitans*, *Flavobacterium*, *Shewanella*, *Corynebacterium*, *Ezakiella*, *Porphyromonas*, *hgcl clade*, *Anaerococcus*, *Rhodobacter*, and *Lactovum* were present in PBM + BSF group than FM fed group. The Fisher Alpha and Chao1 diversity indices of PBM + BSF diet group were significantly ($P < 0.05$) higher than the FM group. The Simpson and Shannon indices were also found to be augmented in PBM + BSF fed marron, however, at higher significance level, α-level of 0.005 and 0.001, respectively, (Table 4). In contrast, only Chao1 diversity index was significantly improved in PBM and Fisher alpha in FM + BSF fed marron, respectively.

## LEfSe based microbial lineages and metabolic pathways in four different groups

The results of LEfSe revealed 21 genus which were significantly modulated in three different dietary groups (FM, PBM, and PBM + BSF) at LDA cut-off value of 8.0 based on Wilcoxon non-parametric *t*-test corrected for multiple hypothesis testing ($P < 0.05$). Out of 21, 16 genera showed to be enriched in PBM + BSF fed marron including species from *Lactobacillus*, *Streptococcus*, *Bacteroidetes*, *Aquabacterium*, *Actinobacteria* etc., (Fig. 3). Significantly ($P < 0.05$) increased bacteria at genus level in PBM fed group were *Aeromonas* and *Limnohabitans*. *Hafnia Obesumbacterium*, *Citrobacter* and *Serratia* were found to be enriched in FM fed marron while no genus was exhibited to be significantly ($P > 0.05$)

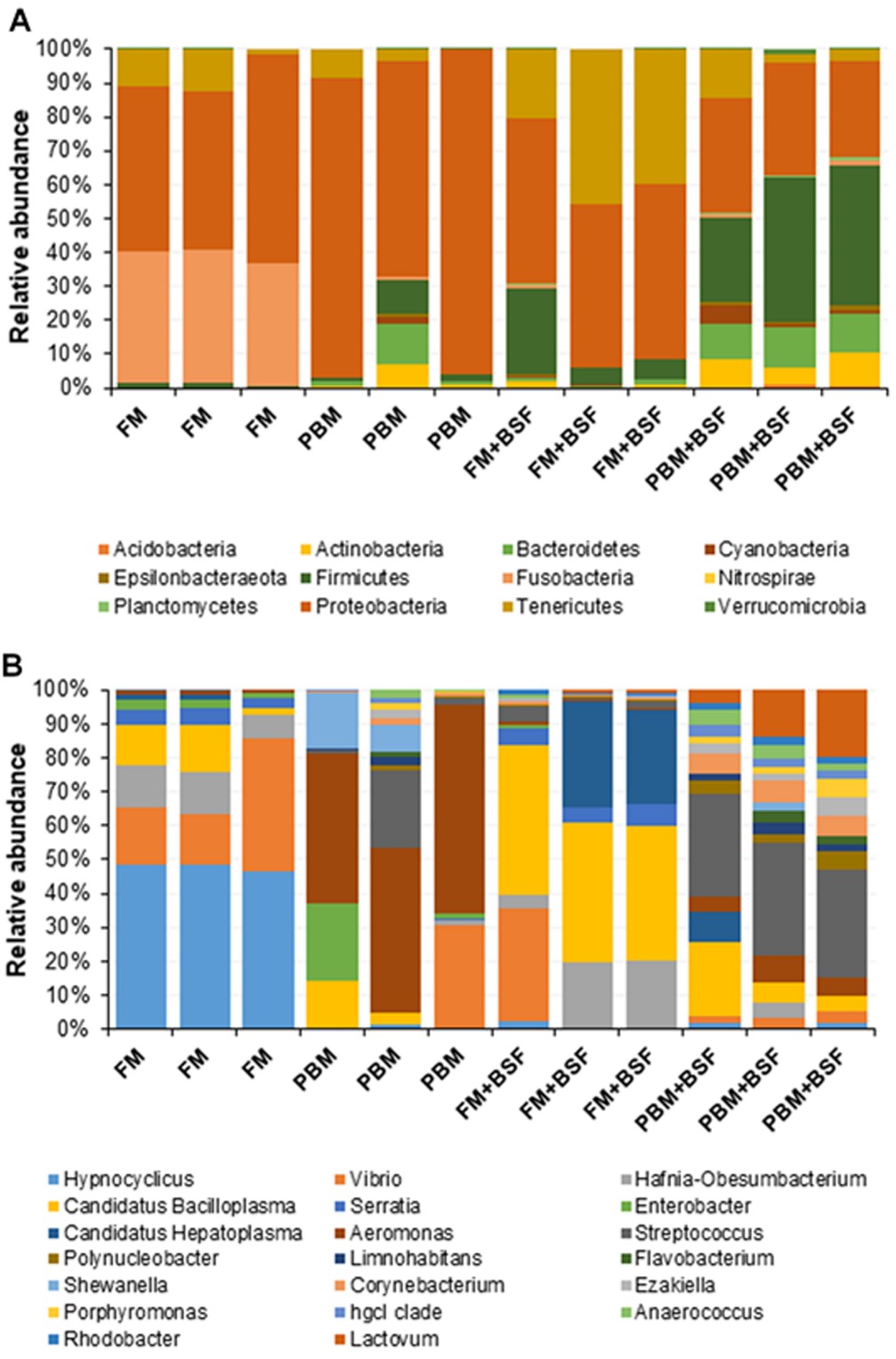

**Figure 2 (A) Relative abundance of bacterial OTUs at phylum level. (B) Relative abundance of bacterial OTUs at a genus level.** Abbreviation: FM, Fishmeal; PBM, Poultry-by-product meal; FM + BSF, Fishmeal + Black soldier fly meal; PBM + BSF, Poultry-by-product meal + Black soldier fly meal.

**Table 4 Major diversity indices (Mean ± SE) of bacterial genera in marron gut at the end of feeding trial.**

| Treatment | Shannon (SE) | Simpson (SE) | Fisher alpha (SE) | Chao1 (SE) |
|---|---|---|---|---|
| FM | 1.51 (0.13) | 0.69 (0.03) | 7.59 (1.36) | 86.74 (10.33) |
| PBM | 1.77 (0.36) | 0.69 (0.07) | 11.34 (1.30) | 116.68 (4.6)* |
| FM + BSF | 1.89 (0.27) | 0.76 (0.04) | 12.3 (1.28)* | 103.99 (3.08) |
| PBM + BSF | 3.19 (0.06)*** | 0.82 (0.006)** | 16.7 (1.91)* | 128.48 (6.2)* |

Notes:
FM, fishmeal; PBM, poultry-by-product meal; FM + BSF, fishmeal + black soldier fly meal; PBM + BSF, poultry-by-product meal + black soldier fly meal.
* Significantly different at α-level of 0.05.
** Significantly different at α-level of 0.005.
*** Significantly different at α-level of 0.001.

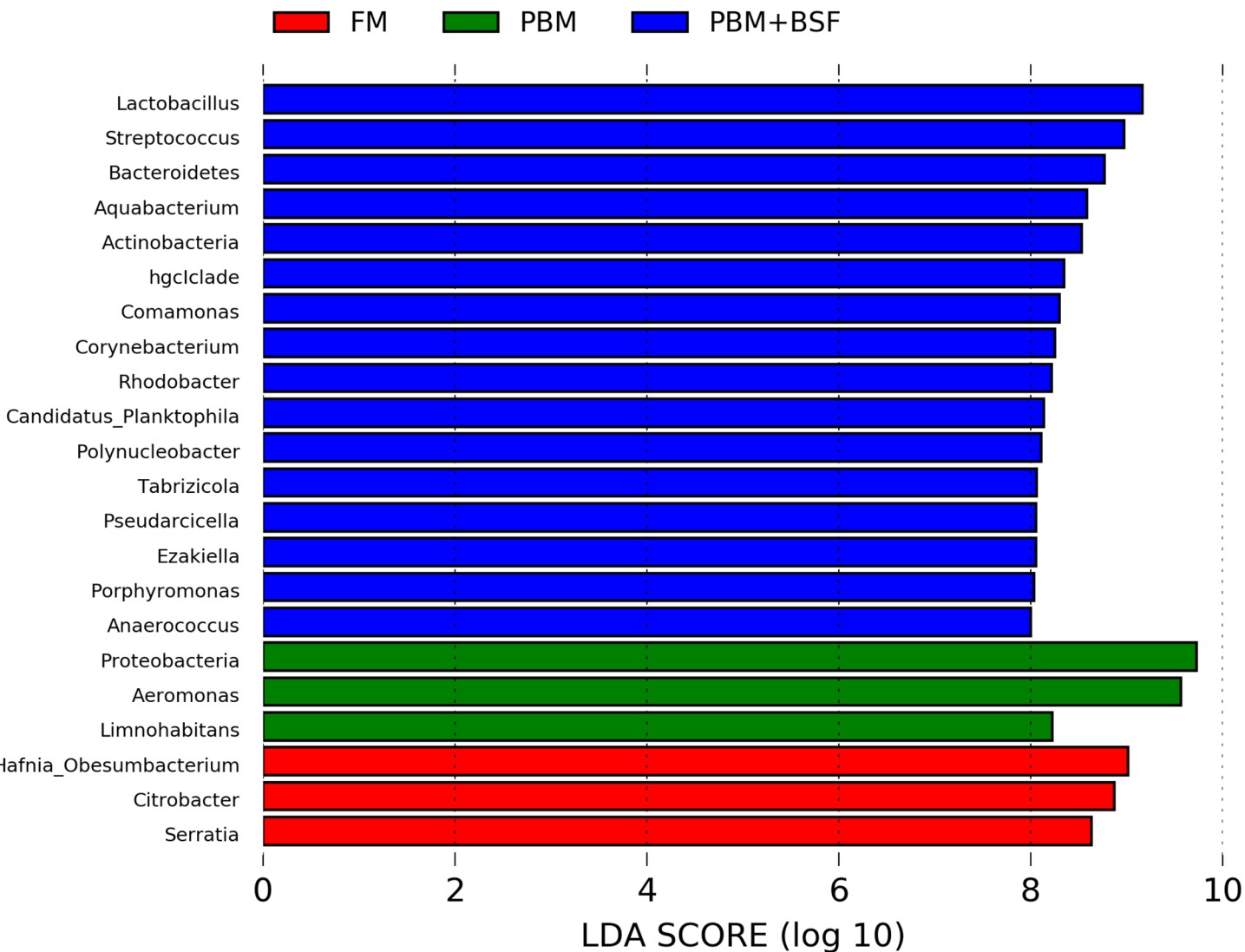

**Figure 3 Indicator bacteria at genus level in three different proteins fed marron groups.** FM + BSF had no significant indicator at LDA cut-off value of 8.0. Abbreviation: FM, Fishmeal; PBM, Poultry-by-product meal; FM + BSF, Fishmeal + Black soldier fly meal; PBM + BSF, Poultry-by-product meal + Black soldier fly meal.

## Cladogram

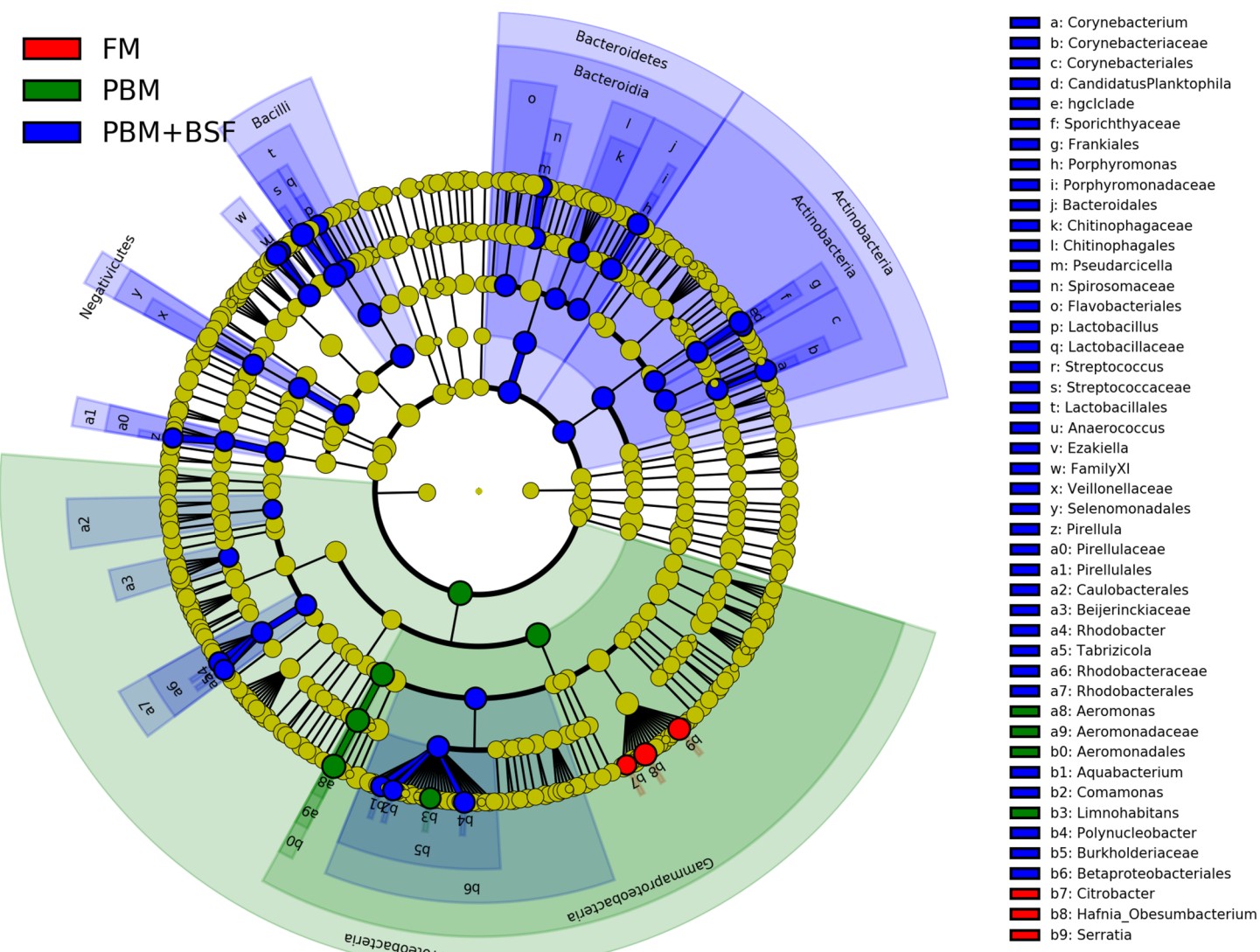

**Figure 4 Circular LEfSe cladogram representing the phylogenetic distribution of bacterial lineage in three different proteins fed marron groups.** The lineage with LDA scores of 3.0 or above are displayed here. The red, green, and blue color indicate FM, PBM, and PBM + BSF, respectively. The dot at center represents the OTUs at phylum level while the outer circle of dots denotes OTUs at genus level. The order, family, and genus that are significantly different between two groups are given in the upper right corner with respective color codes. Abbreviation: FM, Fishmeal; PBM, Poultry-by-product meal; FM + BSF, Fishmeal + Black soldier fly meal; PBM + BSF, Poultry-by-product meal + Black soldier fly meal.

higher in FM + BSF fed marron at stringent cut-off value. LEfSe cladogram revealed significantly (*P* < 0.05) enriched 46 taxa (phylum to genus) in three different dietary groups. No significant enrichment was observed in FM + BSF group at a strict cut-off value (LDA 4.0), where 39 were significantly enriched after fed PBM + BSF diet (Fig. 4). The microbial lineages in PBM + BSF fed marron were exclusively enriched from the phylum *Firmucutes*, *Bacteroides*, and class *Bacilli*, *Negativicutes* whereas in the FM and

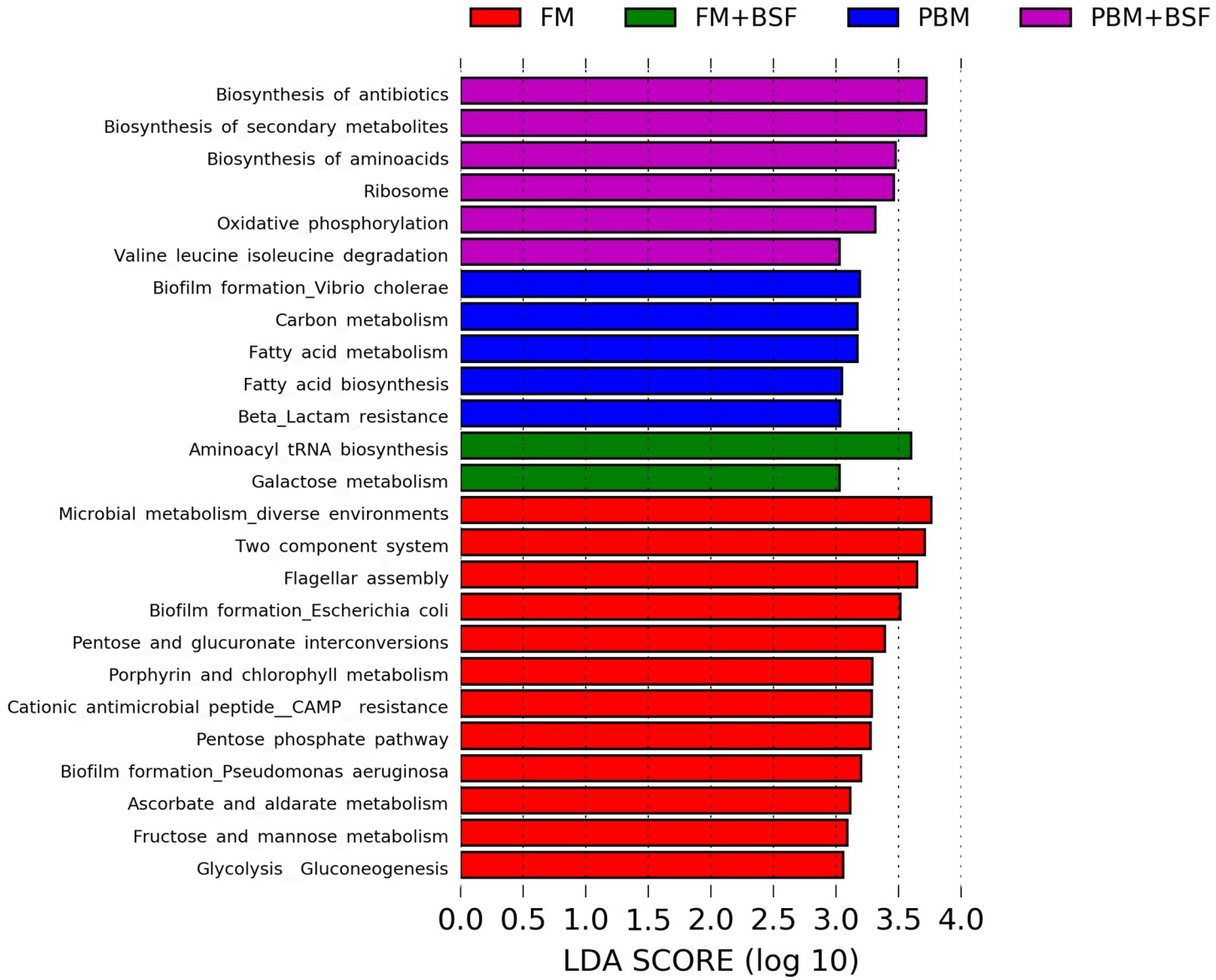

**Figure 5 Predicted differentially expressed KEGG metabolic pathways in four different protein fed marron groups identified by Piphillin and linear discriminant analysis (LEfSe), (LDA > 3.0, P < 0.05).** Abbreviation: FM, Fishmeal; PBM, Poultry-by-product meal; FM + BSF, Fishmeal + Black soldier fly meal; PBM + BSF, Poultry-by-product meal + Black soldier fly meal.

FM + BSF fed group, the enriched lineages were mostly from the *Proteobacteria* phylum. Differences in predicted functional pathways based on the bacterial abundance exhibited to be associated with carbohydrate metabolism and transport among the FM fed marron, fatty acid biosynthesis and metabolism among PBM fed marron, galactose metabolism and aminoacyl tRNA biosynthesis among FM + BSF feed group, and amino acid biosynthesis and energy metabolism in PBM + BSF fed marron (Fig. 5).

## The relative expression level of genes

The relative expression level of seven different genes in four different dietary groups are shown in Fig. 6. Compared to FM dietary group, the BSF supplemented (FM + BSF

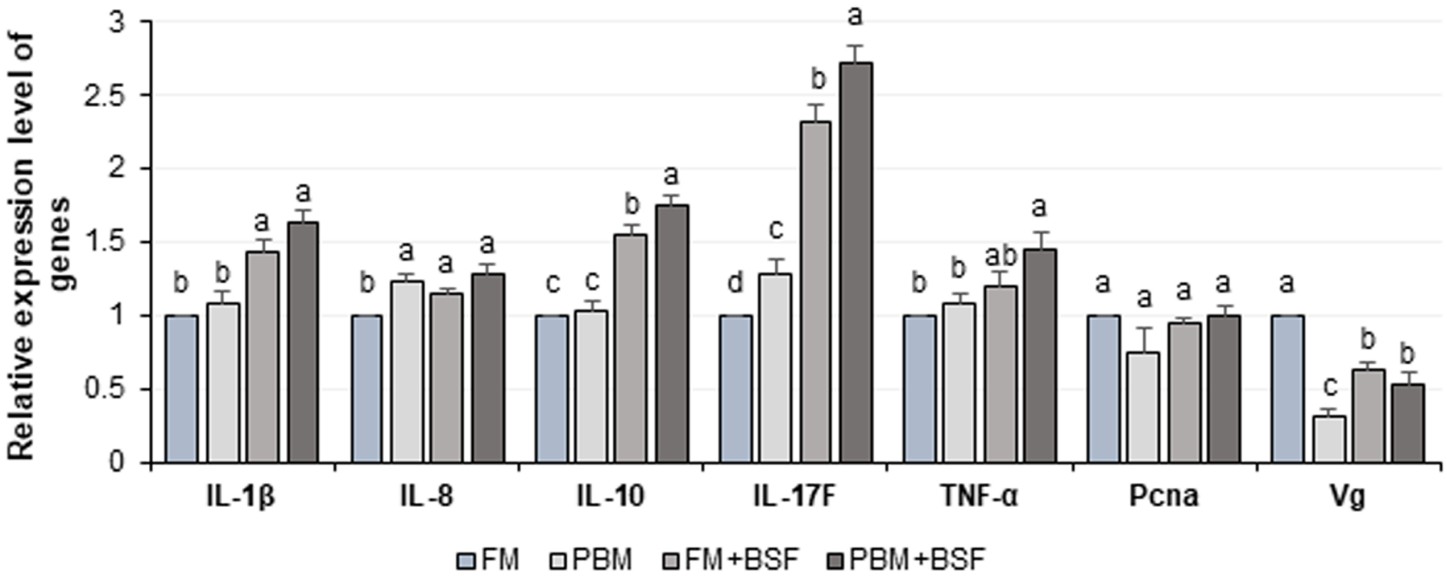

**Figure 6 Relative expression level (Mean ± SE) of major cytokines and crustacean reference genes in marron intestine at the end of 60 days feeding trial.** Means with different superscripts are statistically significant at α-level of 0.05 with Duncan's multiple range test. Abbreviation: FM, Fishmeal; PBM, Poultry-by-product meal; FM + BSF, Fishmeal + Black soldier fly meal; PBM + BSF, Poultry-by-product meal + Black soldier fly meal.

and PBM + BSF) marron gut displayed significant ($P < 0.05$) up-regulation of IL-1β, IL-10, IL-17F, TNF-α, and down-regulation of $Vg$ genes in terms of fold change in contrast to β-actin reference gene after 60 days of feeding trial. The expression of cytokine genes, however, was found to be significantly up-regulated in PBM + BSF fed group. The normalized data showed the highest mean expression for IL-17F, followed by IL-10, IL-1β, and TNF-α, respectively. The relative expression level of IL-8 and $Pcna$ were relatively stable in three different protein fed groups while $Vg$ expression was found to be decreased significantly with PBM and two BSF supplemented diets at the end of the trial.

## DISCUSSION

In aquaculture practice, replacement of dietary FM with alternative cheap protein sources is one of the sought-after research issues that has gained significant momentum in the last couple of decades. Both PBM and BSF have all the ideal properties to be used as a substitute for FM (*Rimoldi et al., 2018*; *Zhou et al., 2018*). To date, several trials have been conducted to analyze the effect of dietary PBM and BSF on the growth performance of aquatic species but yet no research reported the impact on gut microbial structure and cytokine gene expression patterns of commercially important species including crayfish (*Badillo, Herzka & Viana, 2014*; *Gajardo et al., 2017*; *Zhou et al., 2018*). Therefore, the information obtained from this study on the effects of PBM and BSF supplemented diets on gut microbiota and intestine tissue genes will add novel findings that will have a vital contribution to our existing knowledge.

The present study found no significant impacts ($P > 0.05$) of the other three diets (PBM, FM + BSF, and PBM + BSF) on growth parameters of marron, compared to FM. A feeding

trial with different concentration of BSF supplemented diet on Jian carp (*Cyprinus carpio* var. Jian) showed a similar results where the growth rate was independent of BSF supplementation (*Zhou et al., 2018*). As the life cycle of marron under farming conditions is longer than other decapods (*Lawrence, 2007*), and the present experiment lasted for only 2 months, hence, no significant growth rate was noticed. There was no significant change in the health indices between FM and PBM fed marron in this study, but the BSF supplemented diets resulted in significant ($P < 0.05$) positive effects on HO, serum lysozyme, protein and THC, and the impact was more pronounced ($P < 0.005$) with PBM + BSF diet. BSF larvae contains high percentages of protein ($\geq 40\%$), and the inclusion of BSF in the diet reported to increase the protein concentration of fish and poultry (*Wang & Shelomi, 2017*). We found positive impacts of BSF on the percentage of protein in the tail muscle, may be BSF has improved protein assimilation efficiency in marron, however, this cannot be compared with any existing publication. In fish and crayfish, very limited data are currently available for the effects of BSF on the immune response of the host species. A study on boiler chicken had a significant increase in serum lysozyme and other immune parameters including T-lymphocyte, cell proliferation, and disease resistance after 56 days of feeding on BSF supplemented diet (*Lee et al., 2018*).

The gut microbiota of freshwater fish is dominated by *Proteobacteria*, *Firmicutes*, *Bacteroidetes*, *Actinobacteria*, and *Fusobacteria* (*Huang et al., 2014*; *Michl et al., 2017*). The *Proteobacteria*, in addition to *Firmicutes* and *Bacteroidetes*, comprises 90% of bacteria in the fish gut (*Ghanbari, Kneifel & Domig, 2015*). Like human, the majority of beneficial bacterial species are from *Firmicutes* and *Bacteroidetes* in fish (*Rajilić-Stojanović, Smidt & De Vos, 2007*; *Egerton et al., 2018*). *Firmicutes* and *Bacteroidetes* possess the ability to improve the digestibility and immune status of fish to counteract the effects of pathogenic bacteria (*Costantini et al., 2017*). Dietary supplementation of protein sources, poultry and soybean meal, was reported to have significant effects on the modulation of gut microbiota in fish (*Zarkasi et al., 2016*; *Miao et al., 2018*). In the present study, the BSF supplemented diets showed to have significant effects on the bacterial diversity of marron distal gut. Besides *Proteobacteria*, the results showed the abundance of *Fusobacteria* and *Terericutes* in FM and FM + BSF fed marron while *Firmicutes* was dominant in PBM + BSF marron. The 16S rRNA sequence data of *H. illucens* larvae gut showed to have higher percentage of *Bacteroidetes* ($\geq 50\%$), *Proteobacteria* ($\geq 25\%$), and *Firmicutes* ($\geq 15\%$) (*Bruno et al., 2019*). Dietary protein sources can have positive effects on gut microbiota of fish where *Firmicutes*, *Bacteroidetes*, and *Teniricutes* are positively correlated with growth and immune parameters of fish (*Egerton et al., 2018*; *Mekuchi et al., 2018*; *Miao et al., 2018*). In addition, these bacterial groups also have an influential role in colonizing the beneficial bacteria in the fish gut (*Borrelli et al., 2016*; *Vargas-Albores et al., 2017*; *Wang et al., 2018*). The results of the current study found *Firmicutes*, *Teniricutes*, and *Bacteroidetes* richness in BSF supplemented diets that possibly come from the *H. illucens* larvae and associated with higher growth and immunity of marron. At genus level, all four diets triggered the relative abundance of four different genera, *Hynocyclicus* (FM), *Aeromonas* (PBM), *Candidatus* Bacilloplasma (FM + BSF), and *Streptococcus* (PBM + BSF). *Hypnocyclicus* commonly found in the ocean, but its role

in fish and crayfish gut has yet to be revealed (*Roalkvam et al., 2015*). *Aeromonas* abundance has been reported for freshwater fish and some of *Aeromonas hydrophila*, *A. veronii*, *A, sobria*, and *A, salmonicida* have a potential association with fish diseases (*Foysal et al., 2019a*; *Wang et al., 2018*). *Candidatus* Bacilloplasma identified in large scale from all crustacean species including shrimp, lobster, Chinese Mitten Crab (*Eriocheir sinensis*), a hepatopancreatic symbiont helps isopods to survive in nutrient stress, promotes digestion process and up-regulated the expression of immune genes (*Kostanjsek, Strus & Avgustin, 2007*; *Meziti et al., 2010*; *Chen et al., 2015*, *2017*; *Dong et al., 2018*). The present study found enrichment of LAB in the BSF supplemented marron, especially in PBM + BSF more than 50% of bacteria were recorded from genus *Streptococcus*, *Lactovum*, and *Lactobacillus*. LAB are promising probiotic candidates in aquaculture whose health benefits have been widely validated (*Ringø & Francois-Joel, 1998*; *Balcázar et al., 2008*; *Ringø et al., 2018*).

To analyze the indicator microbial lineages, we applied LEfSe, a tool that can effectively predict high-dimensional biomarkers in different conditions from 16S rRNA sequence data. This tool can precisely provide a biological class explanation to define statistical effects, biological consistency, and effect size estimation of the classified biomarkers (*Huang & Jiang, 2016*). With the aid of 16S rRNA data, we identified diverse bacterial lineages in three different feeding regimes where no biomarker was identified for FM + BSF fed marron. Sixteen out of 21 indicator bacterial lineages from the PBM + BSF fed group indicate the enrichment of bacterial population from the selected genus. Besides LAB, *Bacteroides*, *Aquabacterium* and *Actinobacteria* were also found to be improved with PBM + BSF supplemented diet. *Bacteroides* abundance in fish and human gut have been described widely while a lower level of *Bacteroides* reported to have a strong correlation in the progression of several diseases in human (*Zhou & Zhi, 2016*; *Egerton et al., 2018*; *Wang et al., 2018*). The genus *Aquabacterium* has been identified from water and insect (walking sticks, *Phasmatodea*) gut, however, the exact role of this bacteria is yet to be decoded (*Kalmbach et al., 1999*; *Lin et al., 2009*; *Shelomi et al., 2013*). A study demonstrated the inhibitory role of *Actinobacteria* in the fish gut against nine common fish pathogenic bacteria (*Jami et al., 2015*). The genus *Hafnia* reported from healthy crayfish, *Astacus astacus* (L.) showed to have an association with haemorrhagic disease of brown trout (*Salmo trutta*) (*Orozova et al., 2014*). In addition to significantly enriched energy metabolism pathway in all four dietary groups, the PBM + BSF diet also enhanced the antibiotic and amino acid biosynthesis pathways. Limited resources are currently available on the metabolic capabilities of different feeds in the fish gut. However, a study by *Wang et al., (2017)*, found increased protein concentration in the diet had a positive influence on carbohydrate, protein and energy metabolism. Contrarily, antibiotics provide a frontline defense mechanism against the pathogen, investigated to have beneficial function raised from dietary pectin and inulin supplementation (*Johnson et al., 2015*). The PBM + BSF diet, thus, had a positive impact on pathways associated with amino acid biosynthesis, carbohydrate and energy metabolism, driven by bacteria from *Firmicutes* and *Streptococcus*, as previously reported (*Atasoglu et al., 1998*; *Den Besten et al., 2013*; *Dai et al., 2014*; *Bhute et al., 2017*).

The immune response of crayfish is primarily generated from the immunocompetent cells and mucus of intestinal mucosal membrane (*Lieschke & Trede, 2009*; *Ángeles Esteban, 2012*). Among the factors, IL, interferon, TNF are the major cytokine candidates associated with immunity and inflammation of crayfish (*Goins, 2003*; *Araki et al., 2004*; *Jiang et al., 2015b*; *Calderón-Rosete et al., 2018*). The present study revealed significant ($P < 0.05$) up-regulation of pro-inflammatory cytokine genes (IL-1β, IL-17F, and TNF-α) in BSF supplemented diet groups while similar expression patterns were observed for FM and PBM fed groups. To counteract the adverse effect of overexpressed pro-inflammatory cytokines on intestinal tissues, anti-inflammatory cytokines (IL-10) also need to be up-regulated to neutralize inflammation (*Miao et al., 2018*). Enrichment of *Fimicutes* with dietary protein supplementation reported to have a positive role on cytokine gene expression resulting into the improvement of the immune status of fish (*Panigrahi et al., 2007*; *Selim & Reda, 2015*; *Miao et al., 2018*). The bacteria from *Firmicutes* phylum was found to be abundant in PBM + BSF followed by FM + BSF diet groups, and thus we discovered a link between the expression level of cytokine genes and richness of *Firmicutes*. The improved expression level of cytokine genes was recorded with dietary soybean meal where phylum *Firmicutes*, especially *Bacillus* and *Lactobacillus* reported to be increased (*Miao et al., 2018*). LAB associated with the production of pro and anti-inflammatory cytokines, IL-1β, IL-6, IL-8, TNF-α, and IL-10 in the intestine tissue have been reported (*Ringø et al., 2018*). The results of the present study showed a significant up-regulation of intestinal cytokine genes of marron after 60 days of feeding trial in the BSF supplemented groups. The higher abundance (>50%) of *Firmicutes* and LAB in PBM + BSF fed marron might be associated with overexpression of cytokine genes.

## CONCLUSION

The overall findings of the present study showed that the dietary supplementation of BSF significantly improved the intestinal microbiota, health and immune status of marron. However, compared to the other three diets, significant effects of PBM + BSF were recorded for gut microbiota and cytokine genes. However, the interaction mechanism of BSF supplemented diets with bacterial abundance in the gut and associated factors need to be further investigated.

### Funding

This work is supported by the Australian Government Research Training Program (RTP) on behalf of the Department of Education and Training (No. 19059800-Curtin). The funders had no role in study design, data collection and analysis, decision to publish, or preparation of the manuscript.

### Grant Disclosures

The following grant information was disclosed by the authors:
Australian Government Research Training Program (RTP) on behalf of Department of Education and Training: 19059800-Curtin.

## Competing Interests

The authors declare that they have no competing interests.

## Author Contributions

- Md Javed Foysal conceived and designed the experiments, performed the experiments, analyzed the data, prepared figures and/or tables, authored or reviewed drafts of the paper, approved the final draft.
- Ravi Fotedar conceived and designed the experiments, contributed reagents/materials/ analysis tools, authored or reviewed drafts of the paper, approved the final draft.
- Chin-Yen Tay analyzed the data, contributed reagents/materials/analysis tools, authored or reviewed drafts of the paper, approved the final draft.
- Sanjay Kumar Gupta prepared figures and/or tables, authored or reviewed drafts of the paper, approved the final draft.

## Data Availability

Data is available at the National Centre for Biotechnology Information (NCBI), accession number: PRJNA505066.

## Supplemental Information

Supplemental information for this article can be found online at http://dx.doi.org/10.7717/peerj.6891#supplemental-information.

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
