# Peer review of "Dietary supplementation of black soldier fly (Hermetica illucens) meal modulates gut microbiota, innate immune response and health status of marron (Cherax cainii, Austin 2002) fed poultry-by-product and fishmeal based diets"

_PeerJ, doi:10.7717/peerj.6891_

## Round 0.1 · original submission · Major Revisions

Please provide a point-by-point rebutal letter for all the reviewers' comments, along with your revised manuscript.

Reviewer 1 ·

Basic reporting

The MS conforms to the standard of scientific reporting with clear language, structure, figures and tables, fully supporting the hypothesis made with relevant results and discussion.

Experimental design

The present experiment tries to generate information on marrons which are data deficient commercial species. At the same time the present paper explore the potential of alternate animal protein sources to replace fish meal in fish feed.
Experimental design has been well planned and carried out with ethical standard. The methods have been described clearly for to replicate. However another feed (treatment) comprising FM+PBM+BSF could have made interesting observations. Authors may also cite reasons why BSF has not been included beyond 12% in the diet.

Validity of the findings

The data presented are robust and statistically sound to represent the results.However few more observations are made here.

Rimoldi et al 2018 extracted DNA from faecal matter of the sampled fish, to study the microbial communities. In this experiment DNA was extracted from hindgut of the shellfish marrons. So how does it compare or how much of variation you may speculate in the gut microbial diversity.

It is desirable to discuss with more references related to shellfish like comparing gut microbiota of shellfish like L vannamei in which lot of work have been done.

The paper should shed some more light on nutritive values of BSF to correlate with the results.

Line 342 may be corrected by giving proper references or (53,54) may be removed

Additional comments

It is a very well designed and elaborate work addressing many issues pertaining to sustainable aquaculture. Further studies may explore the large scale production potential of alternate protein sources to make commercially viable option to replace fishmeal in aquafeeds.

Reviewer 2 ·

Basic reporting

In their paper entitled Dietary supplementation of alternative protein sources modulate health status, gut microbiota and innate immune response of marron (Cherax cainii, Austin 2002), Foysal et al. tried to evaluate the effects of insect meal as a supplementary diet to fishmeal and poultry meal on the bacterial communtities of marron freshwater crayfish. Apart from bacterial communities diversity studied with illumine, growth indices were calculated as well as other factors such as lysozyme activity and total haemocyte counts, and the presence and abundance of cytocine genes.
Overall the study is interesting and well described. The English language used is clear although some mistakes are observed (e.g. genus, plural genera).
Introduction is well structured with sufficient references and raw data are supplied.

Experimental design

The research question is well defined and the methods are explicitly explained.
However the design lacks some information and can be misleading. For example nowhere is written how many samples (guts) were used for each treatment and method. I suppose that you used 16 samples for each treatment but as I also implied from your figures you used only three samples/ treatment for the 16S analysis (Figure 1). Am I wrong?
Again this is nowhere mentioned in the text and should be added somewhere.

Validity of the findings

Results are well explained and supported by statistical analysis, however in some cases they are hard to understand because of the bad description of samples (How many for what method?). I would suggest the authors to add one more table or provide more information in the manuscript.
The discussion for Bacterial taxonomy is very interesting, but I would suggest you add some more citations/information for the very interesting group of Ca. Bacilloplasma (Tennericutes) that is solely detected in animals gut systems. There are several citations (Wang et al. 2004,Kostanjsek et al. 2007, Meziti et al. 2010) for closely related species that show the importance of this group in gut systems and try to connect it with feeding.

Additional comments

As you will see I recommend the article for major revision since I believe that the addition of a better explanation for samples handling would make your article easier to read.

---

## Round 0.2 · accepted · Accept

Thank you for efficiently revising your manuscript.

# Reviewer 2 ·

Basic reporting

Foysal et al. replied to my major comments regarding experimental design. Also relevant references were added accordingly.

Experimental design

The authors made their methods more clear with the new additions.

Validity of the findings

Data is sufficient now with the changes authors did.